# Module-wise Adaptive Distillation for Multimodality Foundation Models

**Chen Liang**
Georgia Tech
cliang73@gatech.edu

**Jiahui Yu**
Google Research
jiahuiyu@google.com

**Ming-Hsuan Yang**
UC Merced, Google Research
minghsuan@google.com

**Matthew Brown**
Google Research
mtbr@google.com

**Yin Cui**
NVIDIA Research
richardaecn@gmail.com

**Tuo Zhao**
Georgia Tech
tourzhao@gatech.edu

**Boqing Gong**
Google Research
bgong@google.com

**Tianyi Zhou**
University of Maryland, College Park
tianyi@umd.edu

## Abstract

Pre-trained multimodal foundation models have demonstrated remarkable generalizability but pose challenges for deployment due to their large sizes. One effective approach to reducing their sizes is layerwise distillation, wherein small student models are trained to match the hidden representations of large teacher models at each layer. Motivated by our observation that certain architecture components, referred to as modules, contribute more significantly to the student's performance than others, we propose to track the contributions of individual modules by recording the loss decrement after distillation each module and choose the module with a greater contribution to distill more frequently. Such an approach can be naturally formulated as a multi-armed bandit (MAB) problem, where modules and loss decrements are considered as arms and rewards, respectively. We then develop a modified-Thompson sampling algorithm named OPTIMA to address the non-stationarity of module contributions resulting from model updating. Specifically, we leverage the observed contributions in recent history to estimate the changing contribution of each module and select modules based on these estimations to maximize the cumulative contribution. We evaluate the effectiveness of OPTIMA through distillation experiments on various multimodal understanding and image captioning tasks, using the CoCa-Large model [48] as the teacher model.

## 1 Introduction

Large pre-trained multimodality foundation models have demonstrated remarkable generalizability on a wide range of tasks, e.g., image classification, image-text retrieval, and visual question answering [27, 18, 49, 46, 48]. These models often contain several architectural components (referred to as *modules*), each acquiring knowledge of one or more modalities through pre-training. For example, [27] introduce a dual-encoder architecture composed of an image encoder and a text encoder, which are jointly pre-trained to align an image with the corresponding text. This process equips each encoder with both the unimodal representation power and the crossmodal alignment capability. [48] further add a multimodal decoder on top of the dual-encoder architecture. The decoder learns a joint visual and textual representation and obtains the multimodal understanding knowledge. However, the sizes of these models have reached billions of parameters, and the number of modules will further scale to

37th Conference on Neural Information Processing Systems (NeurIPS 2023).

accommodate new modalities. This poses a significant challenge for deployment in applications with latency and storage constraints.

Layerwise distillation is a powerful approach to compressing large models (i.e., teacher models) into small ones (i.e., student models) with minimal loss of performance [16, 30]. This approach trains a student to match the hidden representations of the teacher at each layer. Given that the teacher's layerwise representations often contain rich semantic knowledge, they can significantly improve the student's generalizability [19, 45, 11]. Many existing works have demonstrated the effectiveness of this layerwise strategy in task-specific distillation, where a student is distilled from a teacher that has been fine-tuned on the target task [38, 17].

In task-specific distillation in multimodality models, however, matching the student's representations with those of the teacher at every layer does not always benefit the student's performance. Prior research has shown that improving the representations of a specific modality (i.e., image, text, or multimodality) tends to yield better fine-tuning results than others [51]. Therefore, we hypothesize that the representations of such a modality may contain more relevant knowledge to the target task. If the student focuses on mimicking these representations, it is more likely to achieve better performance on the target task. To verify this hypothesis, we distill multiple students from a CoCa-Large teacher [48], each only matching the teacher's layerwise representations in a single module (i.e., image encoder, text encoder, or multimodal decoder). As shown in Figure 1, distilling a specific module yields significantly better performance than distilling others. If all modules are distilled with equal weighting, interference from other modules may affect the training of this specific module, thereby compromising the student's fitting on the target task (Figure 3).

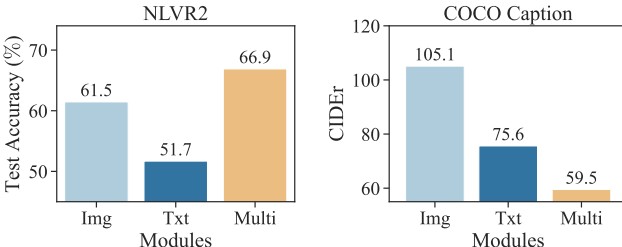

Figure 1: Performances of CoCa-Tiny$_{12}$ students distilled from a CoCa-Large teacher [48] on NLVR2 [37] and Microsoft COCO Caption [6]. Each bar represents the performance obtained by matching only the layerwise representations in a single module. See Details in Section 4.

To mitigate this issue, we propose to track the individual modules' contributions to the distillation process on the target task and choose a module to distill at each step based on their contributions. In this way, we can more frequently distill the module with a greater contribution. To evaluate the contribution of a module, we distill it for a specific number of steps and observe the resulting ratio of decrement in the distillation loss. To track its contribution continually, we repeat this procedure throughout training. Hence, we need to explore different modules by repeatedly evaluating their contributions and, meanwhile, exploit the module with the greatest contribution by distilling it more frequently. Given a limited budget for training steps, we face a significant challenge in balancing exploration and exploitation.

To address this challenge, we adopt a multi-armed bandit (MAB) approach [36, 41]. MAB targets an online decision-making problem that chooses an action (i.e., arm) and observes a reward in each round to maximize the cumulative reward over rounds. MAB algorithms are designed for balancing the evaluation of all arms and choosing the arm that maximizes the cumulative reward, hence can be leveraged to evaluate and choose modules. As illustrated in Figure 2, we consider each module as an arm and every $P$ steps as one round. In each round, we choose a module to distill for $P$ steps and evaluate its contribution as the reward. As more rewards are observed, we obtain a more accurate estimate of the underlying reward distribution for each module, which reflects the actual contribution of this module. Based on the reward distribution, we choose modules to maximize the cumulative loss decrement.

However, the actual contribution of a module may change with model updating. For example, distilling the multimodal decoder may become more beneficial when its input, i.e., the image and text representations, are sufficiently optimized [39]. To estimate such a non-stationary underlying reward

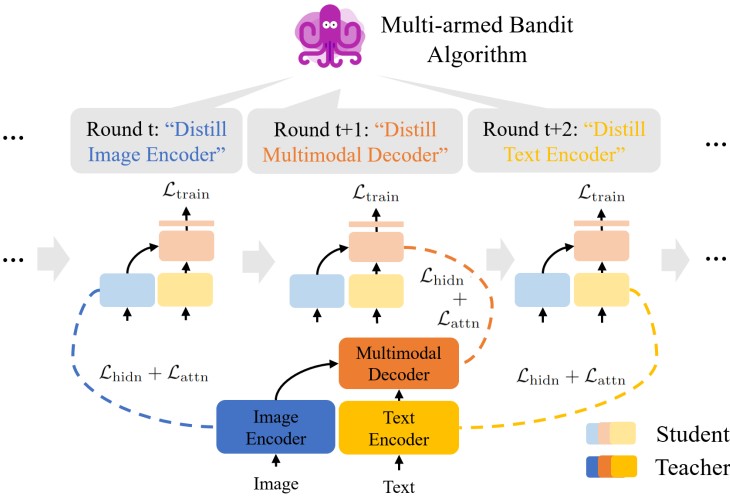

Figure 2: An illustration of OPTIMA.

distribution, we need to count more on rewards observed in *recent history*. This makes it challenging for us to adopt classical MAB algorithms because most of them are designed for stationary underlying reward distributions [4, 10, 40]. Therefore, they *average* the observed rewards over the *whole history* to estimate the underlying reward distribution. In our case, since the model is unlikely to change drastically within a few steps, and so is the underlying reward distribution, we tailor these stationary algorithms by replacing the simple average with the *exponential moving average* of the observed rewards. By discounting old rewards, the reward distribution can track the changing contribution in recent history and provide stable and up-to-date guidance for module selection [13, 15, 28]. By adapting MAB algorithms to the non-stationary environment of multimodality model distillation, we propose OPTIMA (M̲odule-wise Adaptive D̲istillation with M̲ulti-arm B̲andit), a novel task-specific distillation method for multimodality foundation models. We exemplify OPTIMA using the Thompson Sampling algorithm for its strong empirical and theoretical performance [32, 1, 5, 31, 2]. We remark that OPTIMA can be generically combined with other MAB algorithms.

We demonstrate the effectiveness of OPTIMA on various multimodality understanding and image captioning benchmarks [14, 47, 37, 6]. Specifically, we distill a CoCa-Tiny$_{12}$ (102M) and a CoCa-Tiny$_6$ (55M) from a CoCa-Large teacher (672M, [48]), a powerful pre-trained multimodality foundation model. For both students, OPTIMA substantially surpasses the layerwise distillation baseline. Moreover, CoCa-Tiny$_{12}$ outperforms CoCa-Base in three out of four tasks with only $1/3$ its number of layers. Extensive analysis verifies that OPTIMA can track the changing contributions of different modules and choose modules based on their contributions.

## 2 Preliminaries

**Architectures of Multimodality Foundation Models** can be generally categorized into dual-encoder [27, 18, 49, 23], encoder-decoder [42, 46, 44], and Contrastive Captioners (CoCa, [48]). Dual-encoder models contain an image encoder and a text encoder. The two encoders take images and texts as inputs, respectively, and are jointly pre-trained to align an image and relevant text. This enables each encoder to learn both the unimodal representations and the crossmodal alignment knowledge. Encoder-decoder models contain a multimodal decoder on top of an image encoder. The decoder learns a joint visual and textual representation and obtains the multimodal understanding knowledge. To integrate the knowledge from the image encoder, the text encoder, and the multimodal decoder, CoCa combines all modules in a single Transformer and trains them jointly. Specifically, the multimodal decoder takes the representations generated by both the unimodal encoders and learns multimodal representations with the cross-attention mechanism.

**Knowledge Distillation** trains a small model (i.e., student model) to match the output prediction of a large and well-trained model (i.e., teacher model) by minimizing their prediction discrepancy [16]. Specifically, we denote the teacher model as $f_t(\theta_t)$ and the student model as $f_s(\theta_s)$, and consider the

following optimization problem:

$$\min_{\theta_s} \mathcal{L}_{\text{train}}(\theta_s) + \mathcal{D}_{\text{KL}}(\theta_s, \theta_t), \tag{1}$$

where $\mathcal{L}_{\text{train}}(\theta_s)$ is the prediction loss on the target task, e.g., the cross-entropy loss for a classification task; $\mathcal{D}_{\text{KL}}(\theta_s, \theta_t)$ is the KL-Divergence between the probability distributions over their predictions, i.e., $\text{KL}(f_s(\theta_s) \| f_t(\theta_t))$. In large Transformer-based models, distilling knowledge from only the output predictions neglects the rich semantic knowledge in the intermediate layers. To leverage such knowledge, researchers have proposed to match the hidden representations and attention scores at each layer of the teacher and the student [30, 19, 11, 45].

**Thompson Sampling** (TS) is a widely-used MAB algorithm with Bayesian assumption [32, 40, 31, 5, 1]. TS assumes a *prior distribution* on the parameters of the underlying reward distribution for each arm. To estimate the underlying reward distribution, TS updates a *posterior reward distribution* (often referred to as *reward distribution*) for each arm based on the statistics of the observed rewards in history. TS draws an arm from the posterior in each round to maximize the cumulative rewards in the long term. A typical example of TS is to set both the prior distribution and the reward distribution to be Gaussian distributions. In this case, the reward distribution of an arm is specified as

$$\mathcal{D} := \mathcal{N}\left(\mu, \frac{1}{n+1}\right), \tag{2}$$

where $\mu$ is often set as the average of the observed rewards in history and $n$ is set as the number of rounds the arm has been chosen.

## 3 Method

**Problem Formulation.** We consider both the teacher and the student models to be multimodality Transformers containing $c > 1$ modules. For example, they are CoCa models containing $c = 3$ modules in this work: an image encoder, a text encoder, and a multimodal decoder.

We construct $K = 2^c - 1$ arms, each associated with a unique, non-empty subset of modules selected from the $c$ modules. The $k$-th arm is formed as a set of model layers, denoted as $S_k$, in its associated subset. We set $T = \frac{T'}{P}$ rounds, where $T'$ is the total number of training steps and $P \geq 1$ is the number of steps within each round. In this way, choosing the $k$-th arm in the $t$-th round is equivalent to distilling $S_k$ for $P$ steps in the range of $((t-1)P, t \cdot P]$.

For each arm $k \in [K]$, we denote its reward distribution as $\mathcal{D}_k$ and set it as a Gaussian distribution specified as $\mathcal{N}(\mu_k, \frac{1}{n_k+1})$ following Eq. 2. To set a proper prior, we first randomly choose each arm for $T_0 \geq 0$ rounds[1], then initialize $\mu_k = \mu_k^0$ and $n_k = T_0$, where $\mu_k^0$ is the average of the observed rewards over $T_0$ rounds. We explain why setting the reward distribution as a Gaussian distribution is feasible in Appendix A.1.

*Remark 3.1* We consider all possible combinations of modules because, at a certain training stage, there might exist a combination such that distilling it leads to a higher reward than distilling any single module.

*Remark 3.2* Since the dependencies among modules are unclear, we should assume all arms to be dependent and model the reward distribution as a multivariate Gaussian over all arms. However, sampling from such a distribution requires decomposing the covariance matrix, which is computationally costly. To improve efficiency, we ignore the dependency assumption. In practice, we observe no clear disadvantage in the model performance.

**OPTIMA Algorithm.** We initialize

$$\mu_k = \mu_k^0, \quad n_k = T_0 \quad \text{and} \quad \mathcal{D}_k = \mathcal{N}\left(\mu_k, \frac{1}{n_k+1}\right)$$

for all arm $k \in [K]$.

At the $t$-th round, we sample a reward $\hat{r}_k$ for the $k$-th arm from its reward distribution $\mathcal{D}_k$ for all $k \in [K]$. The arm with the highest sampled reward is then chosen for this round, denoted as $a_t$. In this way, the arm is chosen according to the belief of it being the best arm.

---

[1] $T_0$ is included in $T$.

We then play the $a_t$-th arm by distilling $S_{a_t}$ for $P$ steps. At any step $t' \in ((t-1)P, tP]$, we first compute the distillation losses on $S_{a_t}$. Specifically, we denote the hidden representations at the $i$-th layer of the teacher and the student as $H_t^i \in \mathbb{R}^{|x| \times d_t}$ and $H_s^i \in \mathbb{R}^{|x| \times d_s}$, where $d_t$ and $d_s$ denote the hidden dimension and $|x|$ denotes the sequence length. Then the distillation loss on hidden representations is defined as[2]:

$$\mathcal{L}_{\text{hidn}}^{a_t}(\theta_s^{(t')}, \theta_t) = \frac{1}{|S_{a_t}|} \sum_{i \in S_{a_t}} \text{MSE}(H_t^i, H_s^i W_{\text{hidn}}^i), \tag{3}$$

where $\text{MSE}(\cdot, \cdot)$ is the mean-squared error, and $W_{\text{hidn}}^i \in \mathbb{R}^{d_s \times d_t}$ is a randomly initialized and learnable linear projection that projects $H_s^i$ into the same space as $H_t^i$. Similarly, the distillation loss on attention scores is defined as[3]:

$$\mathcal{L}_{\text{attn}}^{a_t}(\theta_s^{(t')}, \theta_t) = \frac{1}{|S_{a_t}|} \sum_{i \in S_{a_t}} \text{MSE}(A_t^i, A_s^i), \tag{4}$$

where $A_t^i, A_s^i \in \mathbb{R}^{|x| \times |x|}$ are the attention score matrices averaged over the multiple attention heads at the $i$-th layer. The student model is then optimized based on a weighted sum of all distillation losses on $S_{a_t}$, i.e.,

$$\mathcal{L}_{\text{total}}^{a_t} = \mathcal{L}_{\text{train}} + \alpha_1 \mathcal{D}_{\text{KL}} + \alpha_2 \mathcal{L}_{\text{hidn}}^{a_t} + \alpha_3 \mathcal{L}_{\text{attn}}^{a_t}, \tag{5}$$

where $\mathcal{L}_{\text{train}}$ and $\mathcal{D}_{\text{KL}}$ are defined in Eq. 1 and $\alpha_1, \alpha_2, \alpha_3 \geq 0$ are hyper-parameters. Specifically, the student model is updated with an SGD-type algorithm [21]:

$$\theta_s^{(t'+1)} \leftarrow \theta_s^{(t')} - \eta \nabla_{\theta_s} \mathcal{L}_{\text{total}}^{a_t}(\theta_s^{(t')}, \theta_t),$$

where $\eta > 0$ is the learning rate. The same procedure is then repeated for $P$ steps.

We then compute the reward $r^{a_t}$ for playing the $a_t$-th arm. Specifically, we design the reward as the averaged ratios of decrements over all types of distillation losses:

$$r_{a_t} = \frac{1}{|U|} \sum_{\mathcal{L}(\cdot) \in U} \max \left( 0, \frac{\mathcal{L}(\theta_s^{((t-1)P)}) - \mathcal{L}(\theta_s^{(tP)})}{\mathcal{L}(\theta_s^{((t-1)P)})} \right). \tag{6}$$

where $U = \{\mathcal{D}_{\text{KL}}(\cdot), \mathcal{L}_{\text{hidn}}(\cdot), \mathcal{L}_{\text{attn}}(\cdot)\}$. Note that $\mathcal{L}_{\text{hidn}}(\cdot)$ and $\mathcal{L}_{\text{attn}}(\cdot)$ are defined in Eq. 3 and Eq. 4, respectively, but by replacing $S_{a_t}$ with the set of all layers. They measure the change in the distillation loss on the *full model*; otherwise, the reward definition changes with the pulled arm and thus cannot provide an evaluation metric consistent across all arms.

The numerator is the difference between $\mathcal{L}(\theta_s^{((t-1)P)})$ and $\mathcal{L}(\theta_s^{(tP)})$. This loss change is accumulated for $P$ steps to reduce the uncertainty caused by noises in the stochastic gradients. As a result, it can reliably reflect the contribution of $S_{a_t}$ to the distillation performance of the full model.

We consider both the changes in output prediction discrepancy ($\mathcal{D}_{\text{KL}}(\cdot)$) and the layerwise representation distances ($\mathcal{L}_{\text{hidn}}(\cdot)$ and $\mathcal{L}_{\text{attn}}(\cdot)$). This improves the robustness of the reward over different tasks because a specific distance metric may not consistently well reflect the distillation performance on all tasks (Figure 7).

For all types of losses, we compute the ratio of the loss change to the loss in the previous round. Normalizing the loss changes prevents the reward from being biased by the scales of different losses. We further clip each ratio by zero to bound the reward between $(0, 1)$, which is desirable for achieving a good performance guarantee in the Thompson Sampling (TS) algorithm. Finally, we average the rewards over all types of losses.

After computing the reward $r_{a_t}$, we update the reward distribution $\mathcal{D}_{a_t}$. Specifically, we update the distribution mean, $\mu_{a_t}$, as the exponential moving average of the observed rewards in the past:

$$\mu_{a_t} \leftarrow \gamma \mu_{a_t} + (1 - \gamma) r_{a_t}, \tag{7}$$

---

[2]We omit the "$(t')$" superscript on $\{H_s^i, W_{\text{hidn}}^i\}_{i \in S_{a_t}}$ to simplify the notations. We assume the teacher and student have the same number of layers. The case of having different layers will be elaborated in Section 4.3.

[3]We omit the "$(t')$" superscript on $\{A_s^i\}_{i \in S_{a_t}}$ to simplify the notations.

where $\gamma \in (0, 1)$ is a hyper-parameter. We then increase $n_{a_t}$ by one and update $D_{a_t}$ following Eq. 2. Since the exponential moving average discounts the old rewards, $\mu_{a_t}$ tends to reflect the average of the rewards within recent rounds. Recall that the actual contribution of each module changes dynamically. By using $\mu_{a_t}$ as the distribution mean, $\mathcal{D}_{a_t}$ can capture the changing contribution and provides up-to-date guidance for module selection. The complete algorithm is shown in Alg. 1[4].

---

**Algorithm 1** OPTIMA: Module Adaptive Distillation

---

1: **Input**: $T$: the number of total rounds; $P$: the number of steps in each round; $K = 2^c - 1$: the number of arms; $\gamma$: a discounted factor. $\theta_s^{(0)}, \theta_t$: the initialized student and the teacher models.
2: **Output**: $\theta_s^{(T')}$
3: $t' = 0, n_k = \mu_k = 0 \; \forall k = 1, ..., K$.
4: **for** $t = 1, ..., T$ **do**
5:     **for** $k = 1, ..., K$ **do**
6:         Sample a reward for arm $k$: $\widehat{r}_k \sim \mathcal{N}\left(\mu_k, \frac{1}{n_k+1}\right)$.
7:     **end for**
8:     Select arm $a_t \leftarrow \arg\max_k \widehat{r}_k$.
9:
10:     **for** $p = 1, ..., P$ **do**
11:         Compute $\mathcal{L}_{total}^{a_t}(\theta_s^{(t')}, \theta_t)$ following Eq. 5.
12:         $\theta_s^{(t'+1)} \leftarrow \theta_s^{(t')} - \eta \nabla_{\theta_s} \mathcal{L}_{\text{total}}^{a_t}(\theta_s^{(t')}, \theta_t)$.
13:         $t' \leftarrow t' + 1$.
14:     **end for**
15:
16:     Compute $r_{a_t}$ following Eq. 6.
17:     Update $\mu_{a_t}$ following Eq. 7.
18:     $n_k \leftarrow n_k + 1$.
19: **end for**

---

# 4 Experiments

We verify the effectiveness of OPTIMA on popular multimodal understanding and image captioning benchmarks.

## 4.1 Data

We conduct task-specific distillation on three multimodal understanding tasks: visual question answering (VQA, [14]), visual entailment (SNLI-VE, [47]), and visual reasoning (NLVR2, [37]). **VQA** is to answer an input question with a sentence based on an input image. It is formulated as a classification problem with 3129 classes where each class corresponds to an answer. **SNLI-VE** is a three-way classification problem to predict whether an input sentence entails the input image, contradicts it, or is neutral to it. **NLVR2** is a binary classification problem to predict whether an input sentence is true or false about an input image pair. We further train and evaluate the model using the **Microsoft COCO Caption** dataset [6] and the Karpathy-test split, respectively. The task is to describe an input image with a sentence. See Appendix A.2 for details.

## 4.2 Models

We evaluate OPTIMA on CoCa [48]. It is a powerful Transformer-based multimodality foundation model pre-trained on web-scale alt-texts [18] and image-text pairs [50]. Since CoCa consists of an image encoder, a text encoder, and a multimodal decoder, it provides a sufficiently large action space for OPTIMA. Furthermore, the decoder produces multimodal representations, which allows it to be directly fine-tuned without any extra step of multimodal pre-training, commonly needed for dual-encoder models [35, 49, 9].

---

[4]We assume $T_0 = 0$ for simplicity of demonstration.

For each target task, we fine-tune a pre-trained CoCa-Large as the teacher (672M, 48 layers (24/12/12 layers in image/text/multimodal modules), $d_t = 1024$). We then distill a task-specific student from the fine-tuned teacher. Specifically, we consider two architectures: CoCa-Tiny$_{12}$ (101.8M, 12 layers (6/3/3 layers), $d_s = 768$) and CoCa-Tiny$_6$ (54.5M, 6 layers (3/1/2 layers), $d_s = 768$). The layers in each module of the student are initialized with the layers uniformly sampled from the corresponding module of a pre-trained CoCa-Base (293M, 36 layers, $d = 768$).

## 4.3 Implementation Details

**Teacher Initialization.** We fine-tune a pre-trained CoCa-Large as the teacher for each target task. For multimodal understanding tasks, we attach a randomly initialized task-specific linear classifier on top of the multimodal decoder for answer prediction. We fine-tune both the model and the classifier using the cross-entropy loss. For the image captioning task, we directly fine-tune CoCa with the captioning loss [48], i.e., $\mathcal{L}_{\text{cap}} = \sum_{\ell=1}^{|x|} \log P_{\theta_t}(y_\ell | y_{<\ell}, x)$. We follow [48] for all fine-tuning hyper-parameters (See Appendix A.3 for details).

**Task-Specific Distillation.** We fix the fine-tuned teacher and distill a student using OPTIMA on each task. For all tasks, we train the student for $T' = 100$k steps. We use Adafactor with decoupled weight decay [34] as the optimizer with $\beta = (0.9, 0.999)$ and a learning rate of $1 \times 10^{-3}$ with a linear decay schedule. We match the $i$-th layer of a student's module with the $\lceil gi \rceil$-th layer of the corresponding teacher's module, where $g$ is the ratio of the number of layers of the two modules. We randomly initialize $W_{\text{hidn}} \in \mathbb{R}^{d_s \times d_t}$. We set $\alpha_1 = 0$, $\alpha_2 = 1$ and $\alpha_3 = 1 \times 10^{-2}$ for all tasks. For OPTIMA, we set $\gamma = 0.98$, $T_0 = 10$, $P = 100$ and $T = \frac{T'}{P} = 1$k. Full details are deferred to Appendix A.4.

## 4.4 Baselines

**Pre-trained Vision-Language Models (VLMs).** To compare with models with similar scales, we present the fine-tuning performance of existing pre-trained VLMs: UNITER [7], OSCAR [24], ViLT [20], VILLA [12], CLIP-ViL$_p$ [35] and ALBEF [23]. Different from foundation models, VLMs often contain a single backbone, e.g., a 12-layer Transformer. The backbone takes both image features and text embeddings as inputs and learns the multimodal representations. They then use an auxiliary model to predict the input image features, e.g., CLIP-ViL$_p$ uses a pre-trained CLIP encoder [27], ALBEF uses a pre-trained ViT-B/16 [8], and the rest use a pre-trained Faster R-CNN [3]. We exclude methods that outperform the CoCa-Large teacher.

**Multimodality Distillation Methods.** We further compare OPTIMA with existing multimodality distillation methods: MiniVLM [43], DistilVLM [11] and DIDE [45]. Different from OPTIMA: 1) All methods conduct distillation in the pre-training stage. Pre-training distillation significantly improves the student's generalizability but requires much more computational resources than task-specific distillation. 2) All methods consider VLMs as the teachers. VLMs are much smaller than CoCa models, and a small teacher-student capacity gap often improves the effectiveness of distillation [26, 25]. However, VLMs are less generalizable and can only distill students on limited tasks. MiniVLM adopts knowledge distillation on additional unlabeled data (Eq. 1). DistilVLM adopts layerwise distillation. DIDE distills a dual-encoders student from a VLM teacher by aligning cross-modal attentions. We ignore concurrent methods with a teacher that significantly outperforms the CoCa-Large teacher.

## 4.5 Main Result

Table 1 summarizes the evaluation results of layerwise distillation and OPTIMA. We report the median over five random seeds[5]. On both CoCa-Tiny$_6$ and CoCa-Tiny$_{12}$, OPTIMA can achieve consistent and notable improvements upon layerwise distillation over all tasks. The gains are most prominent in NLVR2 and COCO, e.g., the gains on CoCa-Tiny$_{12}$ are $0.9$ and $4.4$, respectively. This may be explained by Figure 3, which shows the contributions of modules in these tasks exhibit high variances and the training of a specific module can impair the training of others. The gain is slightly smaller in CoCa-Tiny$_6$ than CoCa-Tiny$_{12}$. This might be because CoCa-Tiny$_6$ converges slower, and that specific module is less likely to over-fit quickly and interfere with others.

---

[5]The standard deviations are reported in Appendix A.5.

Compared with pre-trained VLMs at the same scale, CoCa-Tiny$_{12}$ (OPTIMA) can achieve similar performance with a smaller size. Compared with CoCa-Base, OPTIMA can achieve notable improvements on three out of four tasks with only $1/3$ its number of layers.

Compared with MiniVLM, DistilVLM and DIDE, OPTIMA can achieve significantly better performance on CoCa-Tiny$_{12}$ and similar performance on CoCa-Tiny$_6$ over all tasks. Recall that these baseline methods use web-scale pre-training data for distillation and use single-backbone VLMs as teachers. In contrast, OPTIMA uses limited target task data and faces a much larger capacity gap, e.g., 5 times larger for CoCa-Tiny$_{12}$. Nevertheless, the performance gaps between CoCa-Tiny$_{12}$ (OPTIMA) and CoCa-Large are comparable to those in the baseline methods. Furthermore, while baseline methods demonstrate gains on limited tasks, OPTIMA shows well-rounded gains on all tasks.

Table 1: The evaluation results of OPTIMA and baseline methods on VQA, SNLI-VE, NLVR2 and COCO Caption. The results of all baselines are reported from the original papers. The inference speedup is computed with respect to the batch-averaged inference time of the corresponding teacher model averaged across all tasks. We present the sizes of backbone parameters because the embedding sizes vary largely across methods depending on the implementation details. All sizes are counted based on the released checkpoints from the authors.

| Method | Teacher | Params. (million) | Inference Speedup | VQA test-std | SNLI-VE test | NLVR2 test-p | COCO CIDEr |
|---|---|---|---|---|---|---|---|
| UNITER-Base [7] | - | 146.5 | - | 72.9 | 78.3 | 77.9 | - |
| OSCAR-Base [24] | - | 146.5 | - | 73.4 | - | - | 137.6 |
| ViLT-Base [20] | - | 85.6 | - | - | 76.4 | 76.1 | - |
| VILLA-Base [12] | - | 146.5 | - | 73.7 | 79.0 | 79.3 | - |
| CLIP-ViL$_p$ [35] | - | 187.7 | - | 74.1 | 79.0 | - | 127.9 |
| ALBEF [23] | - | 241.0 | - | 74.7 | 80.3 | 80.5 | - |
| CoCa-Base [48] | - | 293.1 | - | 69.2 | 83.6 | 80.2 | 126.7 |
| CoCa-Large [48] | - | 672.1 | - | 75.3 | 85.6 | 82.6 | 132.3 |
| MiniVLM$_{12 \times 384}$ [43] | OSCAR-Base | 30.0 | 3.1× | 68.1 | - | - | 115.0 |
| DistilVLM$_{12 \times 384}$ [11] | OSCAR-Base | 30.0 | 3.1× | 69.2 | - | - | 117.1 |
| DIDE$_{12}$ [45] | ViLT-Base | 171.3 | 3.0× | 69.2 | 76.3 | 75.6 | - |
| CoCa-Tiny$_6$ | CoCa-Large | 54.5 | 6.2× | 68.7 | 82.0 | 76.5 | 112.2 |
| CoCa-Tiny$_6$ (OPTIMA) | CoCa-Large | 54.5 | 6.2× | 69.0 | 82.3 | 77.0 | 113.5 |
| CoCa-Tiny$_{12}$ | CoCa-Large | 101.8 | 3.4× | 71.2 | 83.9 | 80.4 | 116.8 |
| CoCa-Tiny$_{12}$ (OPTIMA) | CoCa-Large | 101.8 | 3.4× | 71.6 | 84.3 | 81.3 | 121.2 |

# 5 Analysis

In this section, we verify that OPTIMA improves the student's performance, tracks the actual contributions of all modules, and chooses arms based on their contributions. We further investigate the design of the reward function in Appendix A.6 and study the discounted factor of the non-stationary reward distribution in Appendix A.7.

## 5.1 OPTIMA Improves the Student's Performance

Figure 3 compares the performance of the students obtained by 1) *fixed-arm distillation*: distilling a fixed arm over rounds, 2) *random-arm distillation*: randomly choosing an arm to distill in each round, and 3) OPTIMA. The performance of fixed-arm distillation varies largely across tasks. Distilling an arm that contains more modules than other arms does not always improve the student performance, e.g., "Img+Txt+Multi" achieves $80.4$ while "Img+Multi" achieves $81.1$ in NLVR2. In contrast, OPTIMA demonstrates superiority over all fixed-arm and random-arm distillation baselines. This suggests that OPTIMA can find a better weighting for arm selection for each task.

Figure 4 shows the prediction loss and the output layer prediction discrepancy (i.e., $\mathcal{L}_{\text{train}}$ and $\mathcal{D}_{\text{KL}}$ defined Eq. 1) on the dev set are both lower in OPTIMA than in layerwise distillation. This suggests that the student achieves better generalization performance on both the distillation and the prediction of the target task.

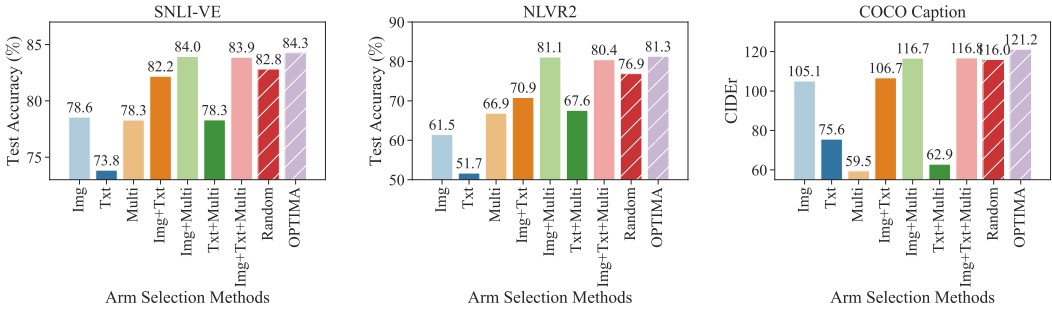

Figure 3: A comparison of the performance of the students by 1) distilling a fixed arm over rounds (shown in solid colors, where each arm is denoted by its associated subset of modules); 2) randomly choosing an arm to distill in each round (denoted by "Random"); and 3) OPTIMA.

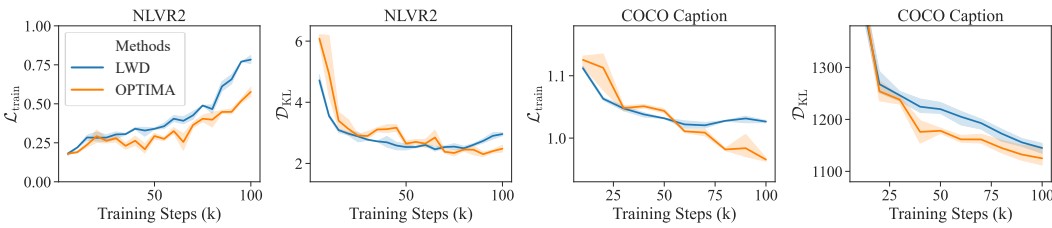

Figure 4: The prediction loss and the output layer prediction discrepancy (i.e., $\mathcal{L}_{\text{train}}$ and $\mathcal{D}_{\text{KL}}$ defined Eq. 1, respectively) in OPTIMA and layerwise distillation (denoted by "LWD").

## 5.2 Reward Distributions Reflect the Contributions of Arms

Figure 5 (Left) shows the means of the reward distributions of all arms through training. In all tasks, "Img+Txt+Multi" quickly dominates the others, while "Multi" and "Txt+Multi" remain incompetent. The reward distributions of all arms slowly evolve through training. For example, in COCO, the arms containing the image encoder all show a non-increasing trend in the later stage of training, while "Txt" shows an increasing trend.

Figure 5 (Right) verifies such evolving reward distributions can correctly reflect the changing contributions of modules. We can observe a strong and positive correlation between the mean of the reward distribution of an arm and the accuracy obtained by distilling this fixed arm through training.

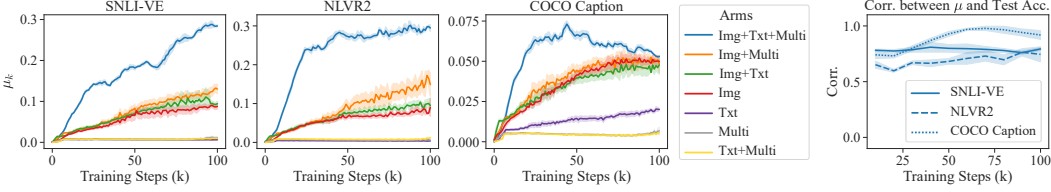

Figure 5: *Left*: The means of the reward distributions of all arms (i.e., $\{\mu_k\}_{k=1}^{K}$) through training. *Right*: The correlation between the means of the reward distributions of individual arms and the test accuracy obtained by distilling these arms through training.

## 5.3 OPTIMA Choose Arms Based on Reward Distributions

Figure 6 visualizes the frequency of choosing each arm through training. In all tasks, we can observe that "Img+Txt+Multi" is the most frequently chosen arm. In COCO, the frequency distributions across arms are flatter than in SNLI-VE and NLVR2, which is consistent with Figure 5 that the means

of the reward distributions are more concentrated across arms. This suggests OPTIMA can choose arms based on the reward distributions.

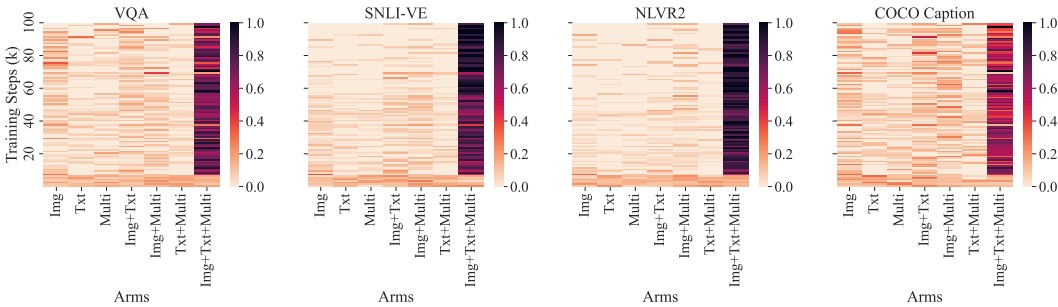

Figure 6: A visualization of the frequency of each arm being chosen through training. A deeper color represents a higher frequency.

# 6 Discussion

**Application to Other Multi-module Multimodality Models.** In this study, we primarily demonstrate the efficacy of OPTIMA when applied to models within the CoCa family. This initial success signifies that OPTIMA could be applicable in scenarios where one or more modules predominantly contribute to the target task, a phenomenon also observed in other multi-module models [35]. The exploration of the application of OPTIMA to other multi-module models is left for future research.

Furthermore, OPTIMA does not incur computational and memory costs exceeding those of layerwise distillation, as it calculates only the necessary layerwise losses and gradients for distilling a subset of modules. This efficiency enables the exploration of OPTIMA's applicability to larger foundational models - those incorporating diverse modules to accommodate not only image and text modalities but also others such as audio and video.

**Design Fine-grained Action Space.** In this study, we divide the CoCa model coarsely into three modules based on modalities, and design the action space to encompass all possible module combinations. However, different layers within a single module could exhibit variable contributions to the target task. For example, lower layers tend to capture high-frequency signals crucial for feature extraction, while upper layers typically learn low-frequency signals, vital for semantic understanding. By further subdividing each module into groups of layers, we can refine the action space to encompass all possible combinations of layer groups, potentially enabling the finding of better arms. Additionally, this enhanced granularity enables the extension of our method to single-module VLMs and single-modality models, such as large language models (LLMs) and vision models.

However, directly applying OPTIMA encounters challenges in this fine-grained scenario. Firstly, the dependencies among layer groups can substantially exceed those among modalities, conflicting with the current sampling strategy which assumes minimal dependencies between arms (See *Remark 3.2* in Section 3). Secondly, expanding the action space prolongs exploration time and, consequently, increases training costs. Given these challenges, adapting OPTIMA to such a fine-grained scenario necessitates modifications in the sampling strategy and explorations into more efficient reinforcement learning algorithms, aspects we designate for future research.

# 7 Conclusion

We propose OPTIMA, a module-wise adaptive distillation method for multimodality models. We employ a simple MAB algorithm to demonstrate that distillation based on the contributions of modalities represents a promising and intriguing research direction. We posit that adopting more sophisticated reinforcement learning algorithms can yield greater improvements, and we designate such explorations for future research.

## 8   Acknowledgements

This project was completed during a full-time internship at Google Research. We thank all collaborators from the Brain team and the Perception team for their insightful feedback and intellectual support. We also extend our thanks to the TPU team for providing abundant computational infrastructure and resources.

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

# A Appendix

## A.1 Gaussian as the Posterior

Here we explain why setting the reward distribution as a Gaussian distribution is feasible. Recall that we consider a non-stationary environment. We design the reward distribution to model the rewards observed in recent history. Within a small number of steps, the model is unlikely to change drastically. Therefore, the observed rewards, which are computed as the ratios of decrement in loss, are unlikely to be skewed. As a result, Gaussian is a feasible choice.

## A.2 Data

For the VQA task, we conduct downstream fine-tuning and testing on the VQA 2.0 dataset [14], which consists of 83k images and 444k questions for training, 41k images, and 214k questions for validation. For the image captioning task on COCO, we use [6] for training and testing. It contains 11k images for training and 5k images for validation and 5k images for testing.

## A.3 Teacher Model Implementation Details

We fine-tune a pre-trained CoCa-Large model [48] as the teacher. It contains 672M parameters in Transformer layers and contains a total of 787M parameters including the embedding size. It contains 24 layers in the image encoder, and 12 layers in the text encoder, and 12 layers in the multimodal decoder. We use a vocabulary size of 64k. We use 576 as the image resolution and 18 as the patch size for image inputs. We use 64 as the max sequence length for text inputs. We follow [48] for fine-tuning hyper-parameters for all tasks, as listed in Table 2.

Table 2: Hyper-parameters for fine-tuning CoCa-Large teacher models.

| Hyper-parameters | VQA | SNLI-VE | NLVR2 | COCO Caption |
|---|---|---|---|---|
| Optimizer | \multicolumn Adafactor with Decoupled Weight Decay | | | |
| Adam $\beta$s | $(0.9, 0.999)$ | | | |
| Gradient Clipping | 1.0 | | | |
| Learning Rate Schedule | Linear Schedule Decaying to Zero | | | |
| Warm-up Steps | 1k | | | |
| Weight Decay Rate | 0.1 | | | |
| Pooler Learning Rate | $5 \times 10^{-4}$ | $1 \times 10^{-3}$ | $5 \times 10^{-3}$ | N/A |
| Encoder Learning Rate | $2 \times 10^{-5}$ | $5 \times 10^{-5}$ | $2 \times 10^{-5}$ | $1 \times 10^{-5}$ |
| RandAugment | 1, 10 | 1, 10 | None | None |
| Training Steps | 100k | 50k | 50k | 50k |
| Batch Size | 64 | 128 | 64 | 128 |
| Dropout of Task Layer | 0.5 | 0.5 | 0.5 | N/A |

For multimodal understanding tasks, we follow [48] to apply an attentional pooler with a single query to extract embedding from the decoder output, and train a linear classifier on top of the pooled embedding. For NLVR2, we construct two input sequences, each containing the concatenation of the description and one image. The two output representations are further concatenated as the input to the classifier. For image captioning, we apply the captioning loss proposed in [48]. We do not use the CIDEr metric-specific optimization [29]. We use a greedy strategy for decoding.

## A.4 Distillation Implementation Details

For each task, we distill a CoCa-Tiny$_{12}$ student and a CoCa-Tiny$_6$ student from a fine-tuned CoCa-Large teacher on that task. CoCa-Tiny$_{12}$ contains 102M parameters in the Transformer layers and contains a total of 152M parameters including the embedding size. CoCa-Tiny$_6$ contains 55M parameters in the Transformer layers and contains a total of 105M parameters including the embedding size. We use 576 as the image resolution and 18 as the patch size for image inputs. To tokenize text input, we use a sentence-piece model [33, 22] with a vocabulary size of 64k trained on the sampled pre-training dataset. We use 64 as the max sequence length for text inputs. We follow [48] for fine-tuning hyper-parameters for all tasks, as listed in Table 3.

We conduct a two-stage distillation for CoCa-Tiny$_6$. We first distill CoCa-Tiny$_{12}$ from CoCa-Large, then use the distilled CoCa-Tiny$_{12}$ as the teacher to teach CoCa-Tiny$_6$. Existing works have shown

that introducing an intermediate-sized teacher reduces the gap between the teacher and the student model, which allows the distillation to be more effective [26].

We distill the model for a total of $T'$ steps, i.e., a total of $T = T'/P$ rounds. Among the $T$ rounds, the first $T_0 \cdot K$ rounds are used for initialization of the parameters of the reward distribution.

Table 3: Hyper-parameters for distilling CoCa-Tiny student models.

| Hyper-parameters | VQA | SNLI-VE | NLVR2 | COCO Caption |
|---|---|---|---|---|
| $T_0$ | | | 10 | |
| $P$ | | | 100 | |
| $T$ | | | 1000 | |
| $\gamma$ | | | 0.98 | |
| $\alpha$s | | $(0.0, 1.0, 1 \times 10^{-2})$ | | |
| Optimizer | | Adafactor with Decoupled Weight Decay | | |
| Adam $\beta$s | | $(0.9, 0.999)$ | | |
| Gradient Clipping | | 1.0 | | |
| Learning Rate Schedule | | Linear Schedule Decaying to Zero | | |
| Learning Rate | | $1 \times 10^{-3}$ | | |
| Warm-up Steps | | 1k | | |
| Weight Decay Rate | | 0.1 | | |
| RandAugment | 1, 10 | 1, 10 | None | None |
| Training Steps ($T'$) | 125k | 100k | 100k | 100k |
| Batch Size | 128 | 384 | 256 | 256 |
| Dropout of Task Layer | 0.5 | 0.5 | 0.5 | N/A |

## A.5 Statistics of Experimental Results

We report the median of five random seeds for experiment results on CoCa-Tiny$_{12}$ and CoCa-Tiny$_6$. Table 4 show the standard deviations of the experimental results in Table 1.

Table 4: The standard deviation of the experimental results in Table 1.

| Method | VQA Acc | SNLI-VE Acc | NLVR2 Acc | COCO Caption CIDEr |
|---|---|---|---|---|
| CoCa-Tiny$_6$ | 0.20 | 0.25 | 0.11 | 0.15 |
| CoCa-Tiny$_6$ (OPTIMA) | 0.22 | 0.13 | 0.35 | 0.15 |
| CoCa-Tiny$_{12}$ | 0.17 | 0.23 | 0.15 | 0.88 |
| CoCa-Tiny$_{12}$ (OPTIMA) | 0.08 | 0.15 | 0.30 | 0.32 |

## A.6 Design of Reward

Recall that we design the reward (Eq. 6) as the averaged ratio of loss decrements over three types of distillation losses: $\mathcal{D}_{\mathrm{KL}}$ (Eq. 1), $\mathcal{L}_{\mathrm{hidn}}$ (Eq. 3) and $\mathcal{L}_{\mathrm{attn}}$ (Eq. 4). Figure 7 compares ours with two variants: 1) $r_{\mathrm{KD}}$: the ratio of loss decrement of $\mathcal{D}_{\mathrm{KL}}$; 2) $r_{\mathrm{LWD}}$: the averaged ratio of loss decrements over $\mathcal{L}_{\mathrm{hidn}}$ and $\mathcal{L}_{\mathrm{attn}}$. We can observe that $r_{\mathrm{KD}}$ performs better than $r_{\mathrm{LWD}}$ in NLVR2 but reversely in COCO. Since captioning tasks often rely more on contextual knowledge in the layerwise representations than classification tasks, the layerwise representation distance may better characterize the distillation performance in COCO. By taking both distance metrics into consideration, $r_{\mathrm{OPTIMA}}$ performs well on both tasks.

## A.7 Design of the Reward Distribution

Recall that we design the mean of the reward distribution (Eq. 7) as the exponential moving average (EMA) of the past rewards. Figure 8 shows a hyper-parameter study on the halflife of the EMA, computed as $-\frac{1}{\log_2 \gamma}$. Halflife is the number of rounds the EMA decays by one-half. We can observe that a too-large or too-small halflife, meaning that counting too many old rewards or counting only instantaneous rewards can both be harmful to the student's performance. This corroborates that the actual contribution of each module is non-stationary in the long term and stationary in the short term, and using EMA with an appropriate $\gamma$ can correctly track the changing contribution.

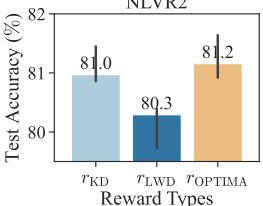 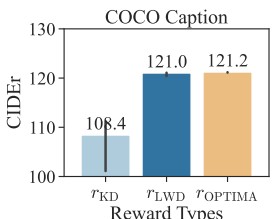

Figure 7: A comparison of three variants of reward: 1) $r_{\mathrm{KD}}$: the ratio of decrement of $\mathcal{D}_{\mathrm{KL}}$; 2) $r_{\mathrm{LWD}}$: the averaged ratio of decrement over $\mathcal{L}_{\mathrm{hidn}}$ and $\mathcal{L}_{\mathrm{attn}}$; 3) $r_{\mathrm{OPTIMA}}$.

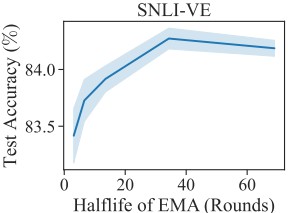

Figure 8: A hyper-parameter study on the halflife of the EMA, computed as $-\frac{1}{\log_2 \gamma}$.

