# OpenReview forum: "Module-wise Adaptive Distillation for Multimodality Foundation Models"
_NeurIPS.cc/2023/Conference — NeurIPS 2023 poster_

### Official Review · Reviewer_3XdA · 2023-07-07

**Soundness:** 3 good
**Presentation:** 3 good
**Contribution:** 3 good
**Rating:** 6
**Confidence:** 3

**Summary:**

The paper proposes a method called OPTIMA for reducing the size of pre-trained multimodal models. It uses layerwise distillation to train smaller student models to match the hidden representations of larger teacher models. The authors track the contributions of individual modules and choose modules with greater contributions to distill more frequently. OPTIMA leverages a modified-Thompson sampling algorithm to estimate changing module contributions and selects modules to maximize cumulative contribution. Experimental results on multimodal understanding and image captioning tasks demonstrate the effectiveness of OPTIMA.

**Strengths:**

The paper works on an important problem: distilling large-scale multi-modal foundation models. The authors explored a relatively new perspective toward this goal, which is Module-wise Adaptive Distillation.  A smart and automatic algorithm, OPTIMA, is proposed to assist in the process, which enabled a balance between the distillation of different model components, defined by which modality (or both) it operates on.  This is overall a new direction for a meaningful topic.

In the presented results, the proposed method outperforms the vanilla layer-wise distillation.

The paper is well written in general. Experimental details are provided, as well as ablation studies on the design choices in OPTIMA.

**Weaknesses:**

The authors did not provide enough evidence/motivation for why OPTIMA is only applied to CoCa. The authors are expected to make the limitation of the proposed method clear. Especially, from the results shown in Table 1, we see that when comparing with other baseline distillation methods, either the DistilVLM12×384 achieves the highest score for the two tasks reported, with the least number of parameters. This leaves doubts about the significance of the work.

Also, the authors chose to report the Median among the 5 repetitions as results in Table 1, which is not a common choice. I'd hope to see a comparison under the mean + std.

**Questions:**

Minor: citations in the Appendix are not compiled correctly.

**Limitations:**

Author provided discussion on the limitation.

---

> ### Author Rebuttal · Authors · 2023-08-10
>
> We appreciate the reviewer for the insightful comments. We hope our response addresses your concerns. Should you have additional questions, we are readily available for further discussion.
>
> **The authors did not provide enough evidence/motivation for why OPTIMA is only applied to CoCa. The authors are expected to make the limitation of the proposed method clear.**
>
> The commonly adopted multimodal foundation models are often computationally expensive to train and test. Given our limited computing resources, we were constrained to allocate them solely to a specific family of models, namely CoCa. Despite this limitation, we present a comprehensive analysis and results focused on CoCa.
>
> Regrettably, we have since lost access to the requisite computing resources, and are unable to provide further results for other model families given the short response/discussion window. While we agree with the reviewers that exploring the application of the module-adaptive idea to other models is of great interest to the community, we need to leave such investigations to future research. This limitation will be further highlighted in our paper.
>
> Nevertheless, CoCa has demonstrated powerful and consistent performance over a diverse set of downstream tasks, thereby implying the broad applicability and efficacy of our method. Furthermore, the improvements on CoCa also indicate the promise of our idea on other multi-module models. We have shown that our idea is effective when a specific module is more useful to a given task than others, which is a shared observation in other multi-module models [1].
>
> [1] Shen, Sheng, et al. "How much can clip benefit vision-and-language tasks?."
>
> **In Table 1, we see that when comparing with other baseline distillation methods, either the DistilVLM12×384 achieves the highest score for the two tasks reported, with the least number of parameters. This leaves doubts about the significance of the work.**
>
> The scores reported by DistilVLM are better than OPTIMA because DistilVLM adopts a different experimental setting from OPTIMA:
>
> As mentioned in line 237-247, 1) DistilVLM conducts distillation in the pre-training stage while OPTIMA conducts distillation in the fine-tuning stage. Pre-training distillation significantly improves the student generalizability but requires much more computations. 2) DistilVLM initializes both the student and the teacher with smaller, but similarly strong models compared with the student and the teacher used in OPTIMA. A smaller teacher-student gap can significantly improve the effectiveness of distillation.
>
> Aside from adopting a different experimental setting, DistilVLM essentially applies layerwise distillation algorithm. As a result, a fair comparison would be between OPTIMA and CoCa-Tiny in Table 1, which is the layerwise distillation baseline under our experimental setting. We can observe that OPTIMA achieves a consistent improvement upon this baseline.
>
> We remark that our main significance is to demonstrate that module-wise adaptive distillation, which distills multi-modality models based on the contributions of modalities, is a promising and interesting research direction. We use a MAB algorithm to demonstrate this point, because MAB is simple, efficient, and good at balancing exploration and exploitation. While we believe that greater improvements can be achieved with more advanced RL algorithms, we will leave the algorithmic explorations for future works.
>
> **Also, the authors chose to report the Median among the 5 repetitions as results in Table 1, which is not a common choice. I'd hope to see a comparison under the mean + std.**
>
> The mean statistics are reported below and are consistent with median results. The std statistics are reported in Table 4 of the paper.
>
> |                  | VQA | SNLI-VE | NLVR2 | CoCo |
> |------------------|-----|---------|-------|------|
> | CoCa-Tiny_6      |  68.7   |    82.0     |    76.5   |   112.1   |
> | CoCa-Tiny_6 (OPTIMA) |   69.0  |  82.3   |   77.0    |  113.3    |
> | CoCa-Tiny_12     |  71.2   |    83.9     |   80.4    |   117.1   |
> | CoCa-Tiny_12 (OPTIMA) |  71.6   |  84.2   |    81.3   |   121.3   |

---

> > ### Author Response · Authors · 2023-08-16
> >
> > Dear Reviewer 3XdA,
> >
> > As we are approaching the midpoint of the discussion period, we would like to cordially inquire about the extent to which we have successfully addressed the concerns outlined in your review. Should there be any lingering points that require further attention, please rest assured that we are enthusiastic about the opportunity to provide comprehensive responses to any subsequent queries or comments you may have.
> >
> > Your constructive input remains invaluable to us, and we appreciate your dedication to enhancing the quality of our manuscript. Thank you for your time and consideration.
> >
> > Best,
> >
> > Authors

---

> > > ### Comment · Reviewer_3XdA · 2023-08-20
> > >
> > > I appreciate the genuine information and the detailed results provided by the authors. The reply does diminish my concern regarding the generalization and significance of the work. In fact, regardless of the scale of the experiments, such work is worth recognization from NeurIPS.  I have no other questions about the paper and will also raise my score.

---

### Official Review · Reviewer_fQyt · 2023-07-07

**Soundness:** 3 good
**Presentation:** 3 good
**Contribution:** 3 good
**Rating:** 5
**Confidence:** 4

**Summary:**

This paper proposes a novel knowledge distillation (KD) method for multimodal foundation models. Motivated by the observation that distilling different modules yields distinct performance, the authors propose to dynamically choose an optimal combination of the modules for KD. Specifically, the choice of the optimal combination is formulated as a multi-armed bandit (MAB) problem. Experimental results demonstrate that the proposed method outperforms other KD baselines.

**Strengths:**

1. The authors introduce an interesting topic on the selection of submodules for KD with multimodal models. Recently, one similar approach has been proposed for distilling image classification models [1].
2. The paper is clearly written and easy to follow.
3. The ablation study appears to be sufficient.
4. The experimental results shows the effectiveness of the propsed method.

**Weaknesses:**


1. **Limitation:** The model architectures adopted in this paper are limited to CoCa-base models. However, several multimodal large models have been proposed recently, such as [2-4]. I believe that if the proposed method can be applied to other foundation models, it will be of great interest to the community. Besides, I am curious to know if the proposed method can be applied to settings where the teacher and the student have different architectures.
2. **Experiment:** What is the training cost of the proposed method compared to other baselines, e.g., training time, GPU memory usage, etc.?

> [1] Song, Jie, et al. "Spot-adaptive knowledge distillation." IEEE Transactions on Image Processing 31 (2022): 3359-3370.
>
> [2] Yang, Ziyi, et al. "i-code: An integrative and composable multimodal learning framework." Proceedings of the AAAI Conference on Artificial Intelligence. Vol. 37. No. 9. 2023.
>
> [3] Lu, Jiasen, et al. "Unified-io: A unified model for vision, language, and multi-modal tasks." arXiv preprint arXiv:2206.08916 (2022).
>
> [4] Shen, Yongliang, et al. "Hugginggpt: Solving ai tasks with chatgpt and its friends in huggingface." arXiv preprint arXiv:2303.17580 (2023).


**Questions:**

The questions are listed in the "weaknesses" section.

**Limitations:**

The authors have discussed the limitations of the proposed method.

---

> ### Author Rebuttal · Authors · 2023-08-10
>
> We appreciate the reviewer for the valuable suggestions. We hope our response addresses your concerns. Should you have additional questions, we are readily available for further discussion.
>
> **The model architectures adopted in this paper are limited to CoCa-base models.**
>
> The commonly adopted multimodal foundation models are often computationally expensive to train and test. Given our limited computing resources, we were constrained to allocate them solely to a specific family of models, namely CoCa. Despite this limitation, we present a comprehensive analysis and results focused on CoCa.
>
> Regrettably, we have since lost access to the requisite computing resources, and are unable to provide further results for other model families given the short response/discussion window. While we agree with the reviewers that exploring the application of the module-adaptive idea to other models is of great interest to the community, we need to leave such investigations to future research. This limitation will be further highlighted in our paper.
>
> Nevertheless, CoCa has demonstrated powerful and consistent performance over a diverse set of downstream tasks, thereby implying the broad applicability and efficacy of our method. Furthermore, the improvements on CoCa also indicate the promise of our idea on other multi-module models. We have shown that our idea is effective when a specific module is more useful to a given task than others, which is a shared observation in other multi-module models [1].
>
> [1] Shen, Sheng, et al. "How much can clip benefit vision-and-language tasks?."
>
> **I am curious to know if the proposed method can be applied to settings where the teacher and the student have different architectures.**
>
> OPTIMA can be easily applied to settings where the teacher and student models have different widths and heights. This can be achieved by modifying the dimension of $W_{hidn}$ and layer mapping function.
>
> The cases where the teacher and student models contain different numbers of modules require further investigation. In such cases, we might need to re-define the arms. For example, if the teacher is a CoCa model and the student is dual-encoder VLM, we may consider the following module mapping as arms: Img->Img, Txt->Txt, Img+Txt->Img+Txt, Img+Multi->Img, Txt+Multi->Txt, Img+Txt+Multi->Img+Txt. Due to the lack of access to computing resources and codebases, we are unable to supplement experimental results under such cases. We will leave the investigations to future works.
>
> **What is the training cost of the proposed method compared to other baselines, e.g., training time, GPU memory usage, etc.?**
>
> OPTIMA incurs a computational cost no more than the layerwise distillation baseline (CoCa-Tiny in Table 1) because OPTIMA only needs to compute the layerwise distillation losses and the corresponding gradients with respect to a subset of modules at each training iteration. OPTIMA does not introduce GPU memory overheads.
>
> As mentioned in line 239-241, all other distillation baselines conduct distillation in the pre-training stage. In contrast, OPTIMA conducts distillation in the fine-tuning stage, which requires much less computational resources.

---

> > ### Comment · Reviewer_fQyt · 2023-08-16
> >
> > Thanks for author's rebuttal. I do not have any further question.

---

> > > ### Author Response · Authors · 2023-08-16
> > >
> > > Thank you for reading our response and providing valuable feedback. If we have addressed your concerns, we would really appreciate if you consider raising your score.

---

> > > > ### Comment · Reviewer_fQyt · 2023-08-18
> > > >
> > > > My main concern is the versatility of the proposed method but the authors fail to provide further results. Therefore, I have to keep my score unchanged.

---

### Official Review · Reviewer_8iuR · 2023-07-09

**Soundness:** 3 good
**Presentation:** 2 fair
**Contribution:** 2 fair
**Rating:** 5
**Confidence:** 4

**Summary:**

This work proposed a new knowledge distillation method for multimodule multimodality models. This work pointed out that different modality modules contribute differently to the model performance, and it proposed to sample different modality modules for distillation during training. The paper proposed a multi-armed badit based method to adaptively select modules during distillation. The method is applied to CoCa model and shows better performance than baselines on multiple datasets

**Strengths:**

1.	The paper is well motivated. It makes sense and is inspiring to show that different modules have different contributions in the knowledge distillation process. Adaptively selecting different modules for distillation could lead to better performance.

2.	The proposed method shows better performance than the baselines. There are thorough ablation study to show the performance of different module selection strategies, which helps the reader to better understand the effectiveness of the method.


**Weaknesses:**

1.	I have some questions about some details. In Table 1, how is the CoCa-Tiny baselines finetuned? Are they finetuned by Image+Text+Multi distillation?

2.	It is hard to find a fair comparison in Table 1. The methods are based on different structures and different teacher models.

3.	The only fair comparison seems to be the comparison with CoCa-Tiny at the third section in Table 1. However, as explained in Q1, one thing is that it is not clear how these beselines are obtained, another is that the improvements seem to be marginal, mostly within 0.5%.

4.	In Figure 3, it also shows that the proposed method is only slightly better than the trivial Image+Multi selection strategy, while being much more complicated.

5.	The method is only applied to CoCa. It would be better to show the effectiveness on more models, such BLIP2.


**Questions:**

Pleae see the weakness part

**Limitations:**

Yes

---

> ### Author Rebuttal · Authors · 2023-08-10
>
> **In Table 1, how is the CoCa-Tiny baselines finetuned? Are they finetuned by Image+Text+Multi distillation?**
>
> Yes, CoCa-Tiny baselines are fine-tuned by “Image+Text+Multi” distillation, where all modules are simultaneously distilled.
>
> **It is hard to find a fair comparison in Table 1. The methods are based on different structures and different teacher models.**
>
> The comparison with CoCa-Tiny at the third section in Table 1 is strictly fair. The comparisons with MiniVLM, DistilVLM and DIDE are not fair. The results are from their original papers, which are achieved under very different experimental settings from ours (e.g., different model architectures and training data).
>
> We have re-implemented MiniVLM to align with our experimental setting[1] and achieved 48.5 on VQA, 72.1 on SNLI-VE, 52.0 on NLVR2 and 79.6 on COCO. We can observe that OPTIMA achieves consistent gains over MiniVLM.
>
> DistilVLM adopts the layerwise distillation method, and the re-implementation of DistilVLM in our experimental setting will become the CoCa-Tiny baseline at the third section in Table 1.
>
> We are unable to re-implement DIDE to align with our experimental setting because DIDE distills a dual-encoder's student from a single-module teacher. The method is architecture-specific and thus cannot use CoCa-Large as the teacher.
>
> We choose to present the results from their original papers because: First, it is practically infeasible to align DIDE with our experimental setting. Second, it is unfair to present the result of a method outside its original experimental setting when that setting is one major contribution of this method. For example, MiniVLM proposed to collect external unlabeled data for distillation, which is one major contribution to its performance. In terms of algorithm, MiniVLM simply adopts knowledge distillation.
>
> Although the results are not fair to compare with, we present these methods because we believe it is important to acknowledge relevant methods. In Section 4.4, we describe their methodologies and summarize their strengths and weaknesses.
>
> [1] First, we replace the teacher and student in MiniVLM with the same teacher and student as in OPTIMA. Second, the original MiniVLM conducts distillation on additional unlabeled pre-training data (Open Images V6) in the vision-language pre-training stage. As our goal is to compare the distillation in the fine-tuning stage, for a better controlled experiment, we only use pre-trained models without training on the unlabelled pre-training data.
>
> **The improvements seem to be marginal, mostly within 0.5%.**
>
> The improvements are both non-trivial and consistent. For example, our experiments on CoCa-Tiny are conducted using five random seeds. Compared with the layerwise distillation baseline (CoCa-Tiny in Table 1), we achieve a median of 0.9 points of gain out of 80.4 on NLVR2 and 4.4 points of gain out of 116.8 on COCO. On VQA with over 3k classes, we improve the accuracy by 0.4 points out of 71.2. Furthermore, we provide analysis to verify that our method works promisingly as designed: the proposed reward can correctly reflect the contribution of the arms, and the arms are chosen based on the reward distribution.
>
> We remark that our major contribution is to demonstrate that module-wise adaptive distillation, which distills multi-modality models based on the contributions of modalities, is a promising and interesting research direction. We use a MAB algorithm to demonstrate this point, because MAB is simple, efficient, and good at balancing exploration and exploitation. While we believe that greater improvements can be achieved with more advanced RL algorithms, we will leave the algorithmic explorations for future works.
>
> **In Figure 3, it also shows that the proposed method is only slightly better than the trivial Image+Multi selection strategy, while being much more complicated.**
>
> “Image+Multi” is only slightly worse than OPTIMA as it is one of the best arms. Although directly distilling the best arm is less complicated than OPTIMA, it requires us to know in advance which arm is the best. Searching for the best arm requires an exhaustive search over all arms because the best arm for one task does not necessarily perform uniformly well on all other tasks. Such an exhaustive search introduces significant complications and computational overheads.
>
> In contrast, OPTIMA is a clean, task-adaptive solution. It requires only a single run (much cheaper than the exhaustive search) and yields a better performance.
>
> **The method is only applied to CoCa. It would be better to show the effectiveness on more models, such BLIP2.**
>
> The commonly adopted multimodal foundation models are often computationally expensive to train and test. Given our limited computing resources, we were constrained to allocate them solely to a specific family of models, namely CoCa. Despite this limitation, we present a comprehensive analysis and results focused on CoCa.
>
> Regrettably, we have since lost access to the requisite computing resources, and are unable to provide further results for other model families given the short response/discussion window. While we agree with the reviewers that exploring the application of the module-adaptive idea to other models is of great interest to the community, we need to leave such investigations to future research. This limitation will be further highlighted in our paper.
>
> Nevertheless, CoCa has demonstrated powerful and consistent performance over a diverse set of downstream tasks, thereby implying the broad applicability and efficacy of our method. Furthermore, the improvements on CoCa also indicate the promise of our idea on other multi-module models. We have shown that our idea is effective when a specific module is more useful to a given task than others, which is a shared observation in other multi-module models [1].
>
> [1] Shen, Sheng, et al. "How much can clip benefit vision-and-language tasks?."

---

> > ### Author Response · Authors · 2023-08-16
> > **Follow-up on the rebuttal**
> >
> > Dear Reviewer 8iuR,
> >
> > As we are approaching the midpoint of the discussion period, we would like to cordially inquire about the extent to which we have successfully addressed the concerns outlined in your review. Should there be any lingering points that require further attention, please rest assured that we are enthusiastic about the opportunity to provide comprehensive responses to any subsequent queries or comments you may have.
> >
> > Your constructive input remains invaluable to us, and we appreciate your dedication to enhancing the quality of our manuscript. Thank you for your time and consideration.
> >
> > Best,
> >
> > Authors

---

> > ### Comment · Reviewer_8iuR · 2023-08-18
> > **Thanks for the rebuttal**
> >
> > Thanks for the detailed rebuttal. Some of my concerns, such as the fair comparison and applicability to other multimodal models, are not fully addressed, but the author explained that they have tried their best to maintain a fair comparison and it is hard to conduct more experiments on other models during rebuttal. I understand the difficultiese. Given that the adaptive multi-module KD could be a promising method, I would like to increase my socre to boarderline accept.

---

> > > ### Author Response · Authors · 2023-08-18
> > >
> > > We thank the reviewer for providing valuable feedback, appreciating our contributions and raising the score!

---

### Official Review · Reviewer_zaxt · 2023-07-21

**Soundness:** 3 good
**Presentation:** 3 good
**Contribution:** 3 good
**Rating:** 6
**Confidence:** 2

**Summary:**

They use layer-wise distillation to reduce model size for multi-modal foundation models. They develop a modified-Thompson sampling algorithm named OPTIMA to address the non-stationarity of module contributions resulting from model updating.

**Strengths:**

The problem is valuable to reduce size for large models. The paper is good in writing. The analysis is interesting.

**Weaknesses:**

The result is based on COCA, experiments on other foundation models are needed. The improvement over baseline is not large.

**Questions:**

No.

**Limitations:**

No.

---

> ### Author Rebuttal · Authors · 2023-08-10
>
> **The result is based on COCA, experiments on other foundation models are needed.**
>
> The commonly adopted multimodal foundation models are often computationally expensive to train and test. Given our limited computing resources, we were constrained to allocate them solely to a specific family of models, namely CoCa. Despite this limitation, we present a comprehensive analysis and results focused on CoCa.
>
> Regrettably, we have since lost access to the requisite computing resources, and are unable to provide further results for other model families given the short response/discussion window. While we agree with the reviewers that exploring the application of the module-adaptive idea to other models is of great interest to the community, we need to leave such investigations to future research. This limitation will be further highlighted in our paper.
>
> Nevertheless, CoCa has demonstrated powerful and consistent performance over a diverse set of downstream tasks, thereby implying the broad applicability and efficacy of our method. Furthermore, the improvements on CoCa also indicate the promise of our idea on other multi-module models. We have shown that our idea is effective when a specific module is more useful to a given task than others, which is a shared observation in other multi-module models [1].
>
> [1] Shen, Sheng, et al. "How much can clip benefit vision-and-language tasks?."
>
> **The improvement over baseline is not large.**
>
> The improvements are both non-trivial and consistent. For example, our experiments on CoCa-Tiny are conducted using five random seeds. Compared with the layerwise distillation baseline (CoCa-Tiny in Table 1), we achieve a median of 0.9 points of gain out of 80.4 on NLVR2 and 4.4 points of gain out of 116.8 on COCO. On VQA with over 3k classes, we improve the accuracy by 0.4 points out of 71.2. Furthermore, we provide analysis to verify that our method works promisingly as designed: the proposed reward can correctly reflect the contribution of the arms, and the arms are chosen based on the reward distribution.
>
> We remark that our major contribution is to demonstrate that module-wise adaptive distillation, which distills multi-modality models based on the contributions of modalities, is a promising and interesting research direction. We use a MAB algorithm to demonstrate this point, because MAB is simple, efficient, and good at balancing exploration and exploitation. While we believe that greater improvements can be achieved with more advanced RL algorithms, we will leave the algorithmic explorations for future works.

---

> > ### Author Response · Authors · 2023-08-16
> >
> > Dear Reviewer zaxt,
> >
> > As we are approaching the midpoint of the discussion period, we would like to cordially inquire about the extent to which we have successfully addressed the concerns outlined in your review. Should there be any lingering points that require further attention, please rest assured that we are enthusiastic about the opportunity to provide comprehensive responses to any subsequent queries or comments you may have.
> >
> > Your constructive input remains invaluable to us, and we appreciate your dedication to enhancing the quality of our manuscript. Thank you for your time and consideration.
> >
> > Best,
> >
> > Authors

---

> > ### Comment · Reviewer_zaxt · 2023-08-20
> > **Thanks for the detailed rebuttal.**
> >
> > Thanks for the detailed rebuttal. Part of my concerns are solved. I will keep my score.

---

### Official Review · Reviewer_1KcH · 2023-08-03

**Soundness:** 3 good
**Presentation:** 3 good
**Contribution:** 2 fair
**Rating:** 5
**Confidence:** 3

**Summary:**

This paper proposes a Module-wise Adaptive Distillation method, OPTIMA, for a multimodal foundation model based on a modified multi-arm bandit algorithm. Comparisons with baseline multimodal pre-trainings and multimodal distillation methods on four different multimodal datasets demonstrate the effectiveness of the proposed method.

**Strengths:**



- the paper is well-written and the idea is clearly presented.
- the idea of using multi-arm bandit to guide distillation is interesting and the effective, as shown in the experiments.

**Weaknesses:**


- Concerning equation 6, I understand that for the sake of fair comparison, the authors choose to compare the loss change of the full model. However, for example, is that reasonable to use the change in $\mathcal{L}_{hidn}$ of "Img" to evaluate the reward of the "Txt" module in the case when we only distill the "Txt" module in the current round.
- the discussion should be elaborated; please look at the Questions below.
- it is mentioned in the introduction that one primary motivation of this paper is that "interference from other modules may affect the training of this specific module." However, later results did not reflect this point. In most cases, the best way is to distill all three modules.

**Questions:**




- In Fig.4, the loss on the dev set of the NLVR2 dataset is increasing, yet the performance of the distilled model is good (Table 1, the performance of CoCa-Tiny12 (OPTIMA) is close to the one of CoCa-Large). Could you provide a more in-depth discussion concerning this interesting behavior?
- In Fig.5, the behavior of rewards seems to be somehow inconsistent with the results in Fig.3. For example, on the NLVR2 dataset, the reward of "Img+Txt" is always higher than "Img"; however, the test accuracy of "Img+Txt" is considerably lower than the one of "Img." Do you have an explanation?
- In this paper, the modules are coarse-grained, corresponding to the individual modality or the fusion. Would considering fine-grained modules be beneficial?

**Limitations:**

Limitations have been addressed in the paper.

---

> ### Author Rebuttal · Authors · 2023-08-10
>
> We appreciate the reviewer for the detailed comments and valuable suggestions. We hope our response addresses your concerns. Should you have additional questions, we are readily available for further discussion.
>
> **Is that reasonable to use the change in $L_{hidn}$ of "Img" to evaluate the reward of the "Txt" module in the case when we only distill the "Txt" module in the current round?**
>
> Pulling one arm will influence the full model performance. Since our goal is to achieve a good full model performance, it is reasonable to design the reward of pulling one arm to reflect its influence on all modules.
>
> In the reviewer’s example, distilling (pulling) “Txt” will change $L_{train}$ and $L_{kd}$. The change will further influence the update of parameters in “Img”, so $L_{hidn}$ of “Img” will change accordingly.
>
> Suppose distilling “Txt” increases $L_{hidn}$ of “Img” (i.e., causes negative interferences to “Img”) and further degrades the full model performance. In such a case, the reward of distilling “Txt” should be decreased as it negatively influences the full model performance. If we fail to take into account the increment in $L_{hidn}$ of “Img” in the reward computation, the reward would remain high. Consequently, the algorithm would keep distilling “Txt”, which may further degrade the full model performance.
>
> **One primary motivation of this paper is that "interference from other modules may affect the training of this specific module." However, later results did not reflect this point.**
>
> We provide an analysis for Figure 3 in lines 272-276: “Distilling an arm that contains more modules than other arms does not always improve the student performance, e.g., “Img+Txt+Multi'' achieves 80.4 while “Img+Multi” achieves 81.1 in NLVR2, …”. This implies that the “Txt” module causes some negative interferences to the training of other modules.
>
> **In Fig.4, the loss on the dev set of the NLVR2 dataset is increasing, yet the performance of the distilled model is good. Could you provide a more in-depth discussion?**
>
> While the prediction loss decreases quickly in the early stage of training (which implies that the student can quickly overfit the final-layer labels), the layerwise distillation loss gradually dominates. Hence, the optimization prefers to minimize the dominating layerwise loss, and therefore reduces the fitting of the student model to the final-layer labels, resulting in an increasing prediction loss.
>
> Layerwise distillation enforces the student to mimic all intermediate-layer semantics of the teacher. Such a strong regularization provides the student with more generalizable knowledge, leading to a good performance in Table 1.
>
> **The reward of "Img+Txt" is always higher than "Img"; however, the test accuracy of "Img+Txt" is considerably lower than the one of "Img.**
>
> We believe that the reviewer misread Figure 3. “Img+Txt” (orange) has a higher accuracy than “Img” (light blue). We will make the presentation clearer in the next version.
>
> **In this paper, the modules are coarse-grained, corresponding to the individual modality or the fusion. Would considering fine-grained modules be beneficial?**
>
> We believe that using fine-grained arms would be beneficial because it may help us find better arm(s) through time. However, this idea requires further investigation. The dependency across arms will increase for fine-grained modules, but the current OPTIMA assumes a low dependency across arms for sampling efficiency. To adapt OPTIMA to more fine-grained arms, alteration on the sampling strategy is needed. Moreover, increasing the number of arms also leads to more expensive exploration costs. We will leave this idea for future research.

---

> > ### Author Response · Authors · 2023-08-16
> >
> > Dear Reviewer 1KcH,
> >
> > As we are approaching the midpoint of the discussion period, we would like to cordially inquire about the extent to which we have successfully addressed the concerns outlined in your review. Should there be any lingering points that require further attention, please rest assured that we are enthusiastic about the opportunity to provide comprehensive responses to any subsequent queries or comments you may have.
> >
> > Your constructive input remains invaluable to us, and we appreciate your dedication to enhancing the quality of our manuscript. Thank you for your time and consideration.
> >
> > Best,
> >
> > Authors

---

> > > ### Comment · Reviewer_1KcH · 2023-08-18
> > > **Thanks**
> > >
> > > Thank you for the detailed reply, which enabled me to appreciate the paper better.
> > > I have no further questions.

---

> > > > ### Author Response · Authors · 2023-08-18
> > > >
> > > > Thank you for reading our responses and providing valuable feedback!  If the rebuttal has addressed your concerns, we would really appreciate it if you would consider raising your score.

---

### Author Response · Authors · 2023-08-13
**Looking Forward to Your Feedback**

Dear reviewers,

We appreciate your insightful reviews and detailed comments. We hope our responses address your concerns. Please feel free to raise further questions or concerns after you read our rebuttal. We hope to catch and answer them during the discussion period.

Thanks,
Authors

---

### Decision · Program_Chairs · 2023-09-21

**Decision:**

Accept (poster)

**Comment:**

Thank you for your submission and the subsequent clarifications provided during the review process.

The paper addresses a significant challenge in the realm of distilling large-scale multi-modal foundation models. The introduction of Module-wise Adaptive Distillation offers a fresh perspective on this topic. The OPTIMA algorithm, as proposed, is both innovative and automatic, effectively striking a balance in the distillation of various model components based on their operational modality. This approach not only presents a novel direction but also underscores the importance of the topic at hand. Furthermore, the superior performance of the method compared to the baselines, coupled with the exhaustive ablation study, provides readers with a deeper understanding of the method's effectiveness.

However, it's worth noting a concern raised by one of the reviewers regarding the fairness of comparisons and the method's applicability to other multi-modal models. While the reviewer acknowledges the challenges you faced and the efforts made to maintain a fair comparison, it would be beneficial to address these concerns in future work or iterations of this research.

Given the overall positive feedback from all reviewers and the constructive discussions that ensued, I am pleased to inform you that your paper has been accepted. We believe that your work will make a valuable contribution to the field and look forward to its presentation.

Area Chair, NeurIPS.